

# Effect of exercise intervention on health-related quality of life in middle-aged and older people with osteoporosis: a systematic review and meta-analysis

Di Geng[1,2], Xiaogang Li[3] and Yan Shi[2]

[1] Department of Physical Education, University of Electronic Science and Technology of China, Chengdu, China
[2] Postdoctoral Research Station in Sports Science, Shanxi University, Taiyuan, China
[3] School of Physical Education, Sichuan Normal University, Chengdu, China

Corresponding author
Yan Shi, tyshiyan@yeah.net

## ABSTRACT

**Background**. Osteoporosis is a common condition affecting middle-aged and older people, posing a serious threat to their health-related quality of life (HRQOL). In recent years, multiple studies have investigated the impact of exercise interventions on HRQOL in middle-aged and older individuals with osteoporosis, but the conclusions have been inconsistent. The aim of this study was to determine the true significance of exercise interventions on HRQOL in middle-aged and older individuals with osteoporosis and to identify optimal exercise prescriptions.

**Methods**. Six databases were searched for RCTs on the impact of exercise interventions on HRQOL in middle-aged and older individuals with osteoporosis. The methodological quality of the study was evaluated with Cochrane risk assessment tool. The effect size pooling, heterogeneity testing, and publication bias were analyzed using Review Manager 5.4 software.

**Result**. A total of 14 RCTs involving 1,214 participants were included, published between 2007 and 2022. The pooled results demonstrated that exercise interventions significantly improved general HRQOL (SMD = 0.79, 95% CI [0.34–1.24], $p = 0.0006$). In terms of physical HRQOL, significant improvements were observed in bodily pain (SMD = 0.51, 95% CI [0.24–0.78], $p = 0.0002$), physical function (SMD = 0.56, 95% CI [0.21–0.91], $p = 0.002$), role physical (SMD = 0.39, 95% CI [0.14–0.64], $p = 0.003$), and general health (SMD = 0.68, 95% CI [0.25–1.11], $p = 0.002$). Regarding mental HRQOL, significant improvements were found in vitality (SMD = 0.58, 95% CI [0.15–1.01], $p = 0.008$), social function (SMD = 0.37, 95% CI [0.17–0.58], $p = 0.0004$), and mental health (SMD = 0.50, 95% CI [0.25–0.74], $p < 0.0001$). Subgroup analysis results indicated that resistance training (SMD = 1.01, 95% CI [0.50–1.52], $p = 0.0001$), intervention frequency of at least three times per week (SMD = 0.80, 95% CI [0.22–1.38], $p = 0.007$), and intervention duration of 13–24 weeks (SMD = 0.85, 95% CI [0.37–1.33], $p = 0.0005$) had large and significant effects on general HRQOL improvements.

**Conclusion**. Exercise interventions improved HRQOL in middle-aged and older individuals with osteoporosis. Resistance training has shown greater benefits than mixed exercises. The optimal frequency is at least three per week, yielding the greatest

improvement. Exercise interventions lasting 13–24 weeks had the most pronounced effect compared to other durations.
**Registration**. PROSPERO (No. CRD42023438771).

## INTRODUCTION

Osteoporosis is a systemic illness that features the decrease in bone mass and the deterioration in the bone tissue microstructure, leading to more brittle bones and higher risks of fractures (*Bijlsma et al., 2012*; *Peyman et al., 2024*). The total number of osteoporosis patients worldwide is estimated at more than 200 million, and the World Health Organization ranks osteoporosis as the second leading cause of death behind cardiovascular disease (*Sözen, Özışık & Başaran, 2017*). Osteoporosis is an age-related bone disease that increases in incidence with age. Middle-aged and older people are at greater risk for osteoporosis, with rates increasing from 24.1% to 51.8% between the ages of 50 and 80 (*Zhu et al., 2022*); osteoporosis increases the risk of hip fracture by a factor of 3–6 (*Kanis et al., 2001*). In the 27 EU countries, there were 27.6 million osteoporosis patients between the ages of 50 and 84 in 2010, and the economic cost of fractures caused by the disease almost reached 37 billion euros (*Hernlund et al., 2013*). China has one of the world's largest aged populations. According to previous epidemiological surveys, approximately 69.44 million people over the age of 50 in China suffer from osteoporosis (*You, 2016*). As the population ages, more individuals are anticipated to develop osteoporosis. Osteoporosis, therefore, has become a serious public health issue posing a threat to the health of middle-aged and older people that deserves sustained attention.

The term "health-related quality of life" (HRQOL) refers to a multifaceted notion that incorporates aspects of one's physical, psychological, social, and mental well-being as well as signs of various diseases (*Treurniet et al., 1997*; *World Health Organization, 2024*). In healthcare systems, HRQOL measurements are frequently used to predict healthcare utilization, compare the relative burden of different diseases, and evaluate the cost-effectiveness of alternative interventions (*Kaplan & Hays, 2022*). A growing body of research has shown that osteoporosis and subsequent fractures have a significant adverse effect on HRQOL. Patients with osteoporosis-induced hip fractures had significantly lower scores on all dimensions of HRQOL within one week of fracture compared to normal controls of the same age; three months after the fracture, the patients' scores on the physiological and social function dimensions were still significantly lower (*Randell et al., 2000*). Even if no fractures occurred, patients with osteoporosis often experienced varied degrees of bodily pain and decreased physical and mental function, which are essential aspects of HRQOL (*Ciubean et al., 2018*). In clinical trials of osteoporosis, HRQOL measurements can provide researchers and clinical physicians with the opportunity to assess

treatment trade-offs and compare the actual effects of different intervention measures in a trial (*Greendale et al., 1993*; *Choo et al., 2024*).

Insufficient physical activity and sedentary behavior have become major causes affecting people's physical and mental health, and are associated with a 6%–10% increase in the incidence of non-communicable diseases including coronary heart disease, type 2 diabetes, breast cancer, and colon cancer (*Lee et al., 2012*). In addition to physiological hazards, these two factors may also contribute to the development of mental disorders (*Goodwin, 2003*). In consideration of health and well-being, World Health Organization recommends that all adults engage in at least 150–300 min of moderate-intensity aerobic activity (or 75–150 min of vigorous activity) per week, among whom older adults should also include balance and coordination activities, as well as strength training (*World Health Organization, 2020*). Currently, there has been abundant research on exercise and the health of older adults. Previous studies have shown that combat sports are able to improve bone health and gait performance in older adults, leading to reduced fall-related injuries (*Ciaccioni et al., 2019*; *Ciaccioni et al., 2020*). Compared to older adults with a sedentary living, those who regularly engage in physical exercise can enhance their functional fitness (flexibility, strength, interlimb coordination, endurance), thereby improving their health perception and life quality (*Ciaccioni et al., 2022*). Although exercise can be an effective strategy to improve health and quality of life for middle-aged and older adults with insufficient physical activity, its effectiveness for patients with osteoporosis is worth discussing. Additionally, there is still controversy regarding the effects of different types of exercise.

In recent years, several studies have investigated the impact of exercise interventions on HRQOL among middle-aged and older patients with osteoporosis, but the conclusions have reached no agreement. For example, one study investigated the effect of 12 weeks of resistance exercise on HRQOL of older patients with osteoporosis, and found that it significantly improved the patients' HRQOL (*Zhang et al., 2022a*). 12 weeks of resistance and balance training had no discernible effects on HRQOL in older osteoporosis patients (*Stanghelle et al., 2020*). In addition, most meta-analysis of the efficacy of exercise interventions on middle-aged and older osteoporosis patients took changes in bone density as primary indicators (*Nikander et al., 2010*; *Zhang et al., 2022b*), whereas HRQOL may improve while the patients' bone density and fracture rate remain the same (*Greendale et al., 1993*). Therefore, the aim of this study was to integrate available evidence to identify the true significance of exercise interventions on HRQOL of middle-aged and older osteoporosis patients and to assess what type of exercise prescription was the best option.

## METHOD

This study is reported according to the Preferred Reporting Items for Systematic Reviews and Meta-Analysis (PRISMA), which can be found in Supplemental Information 1. The protocol for this meta-analysis has been successfully registered with PROSPERO (No. CRD42023438771).

## Search strategy

We systematically searched six databases, including Web of Science, PubMed, Embase, Cochrane Library, China National Knowledge Infrastructure (CNKI), and Wan Fang Database, for RCTs on the impact of exercise interventions on HRQOL among middle-aged and older osteoporosis patients. The time period was set from the establishment of the database to 31 December 2023, and a combination of subject words and free words was used in terms of the retrieval strategy. Using the PubMed database as an example, subjects and free words were used for retrieval: osteoporosis, aged, middle-aged, quality of life, exercise, resistance training, endurance training and randomized controlled trial. See Supplemental Information 2 for the specific retrieval strategies.

## Literature selection

The literature screening criteria was developed according to evidence-based PICOS (participants, intervention, comparison, outcomes, and study design) in medical science (*Liberati et al., 2009*). The studies included must meet the following criteria: (a) participants were clinically diagnosed with osteoporosis (T-score $\leq -2.5$) and aged 45 years or above; (b) exercise interventions were conducted in the experimental group, without limitation on exercise types; (c) the control group did not receive exercise intervention, but maintained their daily activities and/or received routine treatment and health education; (d) the outcome indicators included HRQOL, which was measured by questionnaire; (e) the research adopted a randomized controlled trial as its research design. The exclusion standards included: (a) duplicate literature; (b) articles and abstracts from conferences; (c) books, dissertations and theses; (d) considerable baseline variations observed between the experimental and control groups; (e) studies lack of data for the determination of the effect size; (f) studies not reported in Chinese or English. The literature screening process consisted of three steps: step one, the literature was imported into EndNote X9 literature management software for preliminary processing, and duplicates were removed through automatic and manual methods; step two, the literature that failed to satisfy the inclusion requirements was excluded, which involved title and abstract screening based on the citation information retrieved; step three, the full text of the literature included in the preliminary screening was read one by one to determine whether it should be finally included. Independently, two trained researchers (DG and XL) evaluated the literature in accordance with the inclusion and exclusion criteria. If there was disagreement between the two researchers, which could not be resolved through negotiation, a third researcher (YS) intervened to assist in making a decision. These three reviewers acquire extensive expertise and research experience in the field of sport and exercise science. The study with the most thorough analysis or the biggest sample size was included when data from two or more studies came from the same sample. Interrater reliability for screening was calculated using kappa coefficient (*McHugh, 2012*).

## Data extraction

Two researchers (DG and XL) used an independent double-blind method to collect data from studies that matched the inclusion criteria, entering the data into a pre-made Excel

spreadsheet to minimise errors and potential bias. If the data entered by the two researchers was inconsistent, a third researcher (YS) will ensure it to be re-checked and unified. The content of the data extracted was as follows: (a) fundamental literary information (name of the author, date of publication); (b) basic information of the research object (sample size, age, gender, fracture history); (c) intervention plan (category, time, frequency, period); (d) measurement tools and results of outcome indicator HRQOL. In case of two measurement tools adopted in the study for outcome indicators, the outcome data were extracted respectively.

## Intervention categories and outcome measurement

The categories of exercise interventions are broadly classified as aerobic exercise, resistance training, balance training, multicomponent exercise, *etc.* Aerobic exercise is a continuous form of exercise designed to enhance the body's metabolic function, and includes walking, running, gymnastics, swimming, Tai Chi, and more (*Alves et al., 2015*). Resistance exercise is a form of exercise that increases muscle strength by overcoming external resistance, which can be formed by using elastic bands, dumbbells, barbells, other instruments, or body weights (*Loveless & Ihm, 2015*). Balance training refers to functional training aimed at improving the human body's balance function. It is mostly practiced on a balance board, balance beam, or narrow path by means of walking, body movements, balance exercises, *etc.* Multicomponent exercise is a combination of two or more training methods, typically consisting of aerobic exercise, resistance training, balance training, or other forms of exercise. If the included studies did not clearly report the category of exercise interventions, we classified them based on the criteria described above.

The measurement of HRQOL is mainly achieved through self-evaluation scales. According to different scoring methods, they can be divided into two forms: one is forward scoring scales, such as Short Form 36 Health Survey (SF-36), Osteoporosis Quality of Life Questionnaire (OQLQ), and EuroQol Five Dimensions Questionnaire (EQ5D), a higher score indicates a better HRQOL of the patients; another is reverse scoring scales, such as Quality of Life Questionnaire of the European Foundation for Osteoporosis (Qualeffo-41), where a higher score indicates a poorer HRQOL of the patients. In order to maintain data consistency and facilitate consolidation for effect size calculation, we converted the mean outcome indicators of the experimental group and those of the control group to negative values when reverse scoring scales were used in the studies included.

## Bias risk assessment

Two reviewers independently assessed the methodological quality of the studies included using the Cochrane risk assessment tool in accordance with the principles of evidence-based medical studies (*Sterne et al., 2019*). The Cochrane risk of bias tool was adopted to evaluate the studies' quality in six areas: selection bias (random sequence generation and allocation concealment), performance bias (blinding of participants and staff), detection bias (blinding of outcome assessors), attrition bias (incomplete outcome data), reporting bias (selective reporting of results), and other bias. To express their opinions, reviewers used the terms "low risk of bias", "high risk of bias" or "unclear risk of bias". In case of

inconsistent assessment results, a third researcher was involved in the discussion to reach a consensus, and a bias risk graph was ultimately generated.

## Statistical analysis

Statistical analysis was processed with Review Manager 5.4 software to create forest and funnel plots. Due to the fact that the experimental data was a continuous variable and the measurement tools were different, we utilized standardized mean difference (SMD) and 95% confidence interval (95% CI) as the effect scales to merge the effects. The threshold for statistically significant merge effects is $P < 0.05$. The $Q$-statistic and $I^2$ statistic were used to assess the degree of heterogeneity in the study. The random effects model was adopted for significant heterogeneity ($P < 0.10$, $I^2 \geq 50\%$), whereas the fixed effects model was employed for non-significant heterogeneity ($P \geq 0.10$, $I^2 < 50\%$). In case of significant heterogeneity between studies, a one-to-one exclusion method was used in the sensitivity analysis to explore possible sources of heterogeneity. It should be noted that final measurements of outcome indicators were used when combining the effects, rather than baseline variation values, as not all literature included reported values prior to and after the interventions. In studies where only baseline and change values were available, the research by *Shu et al. (2020)* was referenced, and the final measurements were able to be estimated using the following formula:

$$mean_{final} \approx mean_{base} + mean_{change}, SD_{final} \approx \frac{2 \times R \times SD_{base} + \sqrt{4 \times R^2 \times SD^2_{base} - 4 \times SD^2_{base} + 4 \times SD^2_{change}}}{2},$$
$R = 0.5.$

## RESULTS

### Literature search results

Figure 1 depicts the selection procedure for the literature included in this study. 1,328 articles were obtained by retrieving information from six databases. By importing these articles into literature management software, 469 duplicate articles were removed. Next, to screen the 859 articles remained, their titles and abstracts were read, after which 795 more articles that did not meet the requirements of this study were excluded. Then, by thoroughly reading the full text of the 64 articles that were initially included, 50 articles were finally removed, including those with data concerning duplicate sample ($n = 3$), unavailable data ($n = 8$), control group not eligible ($n = 6$), non-Chinese or English articles ($n = 3$), no outcomes of interest ($n = 6$), ineligible interventions ($n = 7$), ineligible participants ($n = 15$), and unreported baseline data ($n = 2$). In the end, 14 studies in total were incorporated into the meta-analysis (*Alp et al., 2009*; *Arnold et al., 2008*; *Çergel et al., 2019*; *Evstigneeva et al., 2016*; *Ferrara et al., 2019*; *Fu & Fan, 2021*; *Gibbs et al., 2020*; *Hongo et al., 2007*; *Kanemaru et al., 2010*; *Khalili et al., 2016*; *Kronhed et al., 2009*; *Sen, Esmaeilzadeh & Eskiyurt, 2020*; *Stanghelle et al., 2020*; *Zhang et al., 2022a*). There was strong agreement between the reviewers for the screening records and full texts (Kappa: 0.89).

### Basic characteristics of included literature

Table 1 lists the detailed information of the 14 studies included, with all of them published between 2007 and 2022. Next, we describe the features that may be sources of heterogeneity
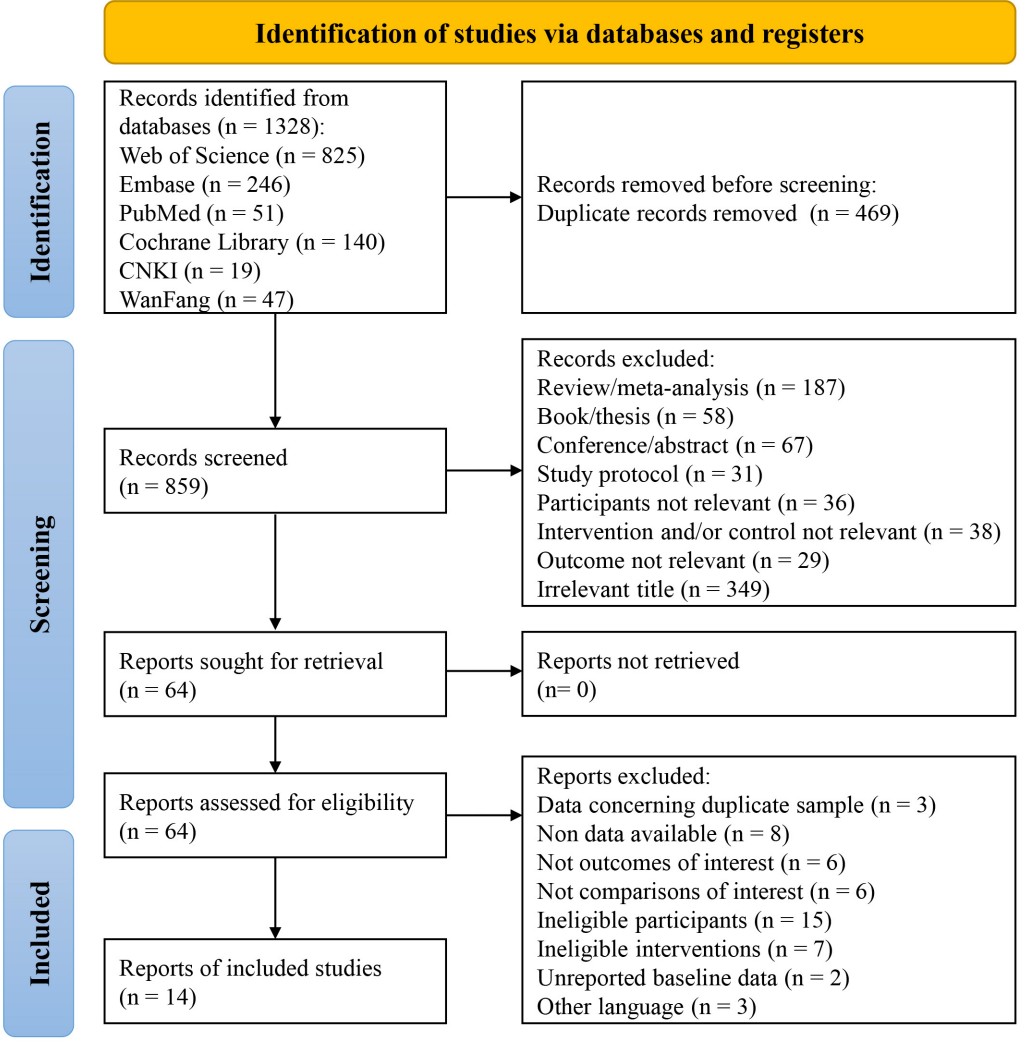

**Figure 1    A flow diagram of the search results and study selection.**

in the results in a brief manner. The sample sizes included in these studies ranged from 34 to 183 people, with a total of 1,214 middle-aged and older osteoporosis patients recruited, the vast majority of whom were women (94.81%, $n = 1,151$). The majority of participants enrolled in six studies had a history of osteoporotic fractures, while participants enrolled in five studies had no such history. The other three studies did not report relevant information explicitly.

The categories of exercise interventions included in this study exhibited significant differences. According to the previously established classification criteria for exercise interventions, they can be divided into three categories: aerobic exercise ($n = 3$), resistance training ($n = 6$), and multicomponent exercise ($n = 5$). In the category of aerobic exercise, two out of three studies used Tai Chi as an intervention, which is known for its low-intensity, balance, and coordination training; the remaining one study implemented personalized

**Table 1 Characteristics of eligible studies.**

| Study | Sample size (male/female) | Age (years) | Fr (Y/N/NS) | Intervention | | | HRQOL scale |
|---|---|---|---|---|---|---|---|
| | | | | Description | Frequency | Duration | |
| Alp et al. (2009) | EG = 22(0/22) CG = 22(0/22) | EG = 69.5 ± 4.9 CG =71.2 ± 6.3 | N | Aerobic exercise (Tai Chi) | 60 min/session, 3 sessions/week | 26 weeks | SF-36 |
| Arnold et al. (2008) | EG = 21(0/21) CG = 27(0/27) | EG = 68.6 ± 5.4 CG = 67.7 ± 6.3 | N | Multicomponent exercise (aerobic, resistance, balance) | 50 min/session, 3 sessions/week | 20 weeks | OQLQ |
| Çergel et al. (2019) | EG = 20(0/20) CG = 20(0/20) | EG = 58.90 ± 4.70, CG = 59.65 ± 6.45 | Y | Resistance exercise (back extensor training) | 3 sessions/week | 6 weeks | Qualeffo-41 |
| Evstigneeva et al. (2016) | EG = 40(0/40) CG = 38(0/38) | EG = 70.7 ± 8.1 CG = 67.6 ± 7.0 | Y | Multicomponent exercise (resistance and balance) | 40 min/sessions, 2 sessions/week | 52 weeks | Qualeffo-41 |
| Ferrara et al. (2019) | EG = 56(0/56) CG = 42(0/42) | EG = 71.61 ± 7.97 CG = 69.71 ± 8.61 | N | Aerobic exercise (Tai Chi) | 45min/sessions, 2 session/week | 26 weeks | SF-36 |
| Fu & Fan (2021) | EG = 63(29/34) CG = 54(23/31) | EG = 63.73 ± 5.64 CG = 62.04 ± 5.33 | NS | Aerobic exercise (running, dancing, swimming, etc.) | 60–90 min/session, 7 sessions/week | 26 weeks | SF-36 |
| Gibbs et al. (2020) | EG = 71(0/71) CG = 70(0/70) | EG = 76.4 ± 6.4, CG = 77.0 ± 7.3 | Y | Multicomponent exercise (aerobic, resistance, balance) | 3 sessions/week | 52 weeks | EQ5D, OQLQ |
| Hongo et al. (2007) | EG = 42(0/42) CG = 38(0/38) | EG = 67 ± 5 CG = 67 ± 7 | NS | Resistance exercises (back extensor training) | 5 sessions/week | 17 weeks | OQLQ |
| Kanemaru et al. (2010) | EG =37(0/37) CG =32(0/32) | EG = 74.7 ± 6.7 CG = 74.3 ± 7.0 | N | Resistance exercise (Multi-muscle training) | 7 sessions/week | 52 weeks | SF-36 |
| Khalili et al. (2016) | EG = 92(0/92) CG = 91(0/91) | EG = 59.40 CG = 59.84 | N | Resistance exercises (back extensor training) | 5 sessions/week | 26 weeks | SF-36 |
| Kronhed et al. (2009) | EG = 31(0/31) CG = 34(0/34) | EG = 69.8–73.4 CG = 69.4–73.0 | Y | Multicomponent exercise (resistance and balance) | 60 min/session, 2 sessions/week | 17 weeks | SF-36, Qualeffo-41 |
| Sen, Esmaeilzadeh & Eskiyurt (2020) | EG = 16(0/16) CG = 18(0/18) | EG = 53.1 ± 4.4 CG = 54.5 ± 6.0 | NS | Resistance exercises (Multi-muscle training) | 20–60 min/session, 3 sessions/week | 24 weeks | Qualeffo-41 |
| Stanghelle et al. (2020) | EG = 76(0/76) CG = 73(0/73) | EG = 74.7 ± 6.1 CG = 73.7 ± 5.6 | Y | Multicomponent exercise (resistance and balance) | 60 min/session, 2 sessions/week | 12 weeks | SF-36, Qualeffo-41 |
| Zhang et al. (2022a) | EG = 34(5/29) CG = 34(6/28) | EG = 68.4 ± 4.9 CG = 68.4 ± 4.6 | Y | Resistance exercise (multi-muscle training) | 45–60 min/session, 3 sessions/week | 12 weeks | SF-36 |

**Notes.**

EG, Experimental Group; CG, Control Group; Fr, Fracture; NS, Not specified; SF-36, Short Form 36 Health Survey; Qualeffo-41, Quality of Life Questionnaire of the European Foundation for Osteoporosis; EQ5D, EuroQol Five Dimensions Questionnaire; OQLQ, Osteoporosis Quality of Life Questionnaire.

exercise programmes based on individual preferences and characteristics, such as running, dancing, hiking, and swimming. In the category of resistance training, six studies were further subdivided: three studies focused on back extensor strength training programmes aimed at enhancing the strength and stability of core muscle groups; and the other three studies adopted strength training programmes involving multiple muscle groups to comprehensively improve muscle strength and function. As for the multicomponent exercise category, three out of five studies combined resistance training and balance training to promote comprehensive physical fitness and prevent falls; the other two studies adopted a comprehensive approach that included aerobic exercise, resistance training, and balance training, aiming to improve the physical health level of participants in a comprehensive manner.
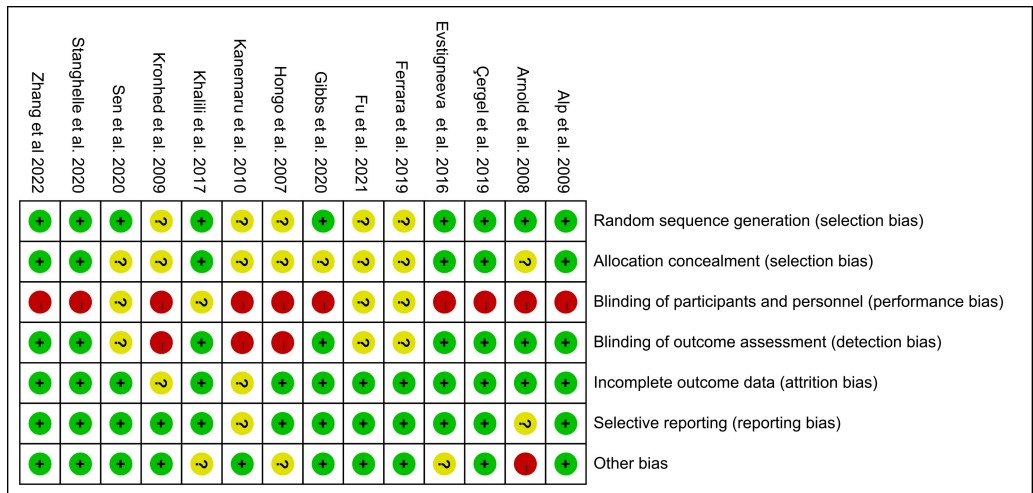

**Figure 2  Risk of bias of included studies.** Note. *Alp et al., 2009*; *Arnold et al., 2008*; *Çergel et al., 2019*; *Evstigneeva et al., 2016*; *Ferrara et al., 2019*; *Fu & Fan, 2021*; *Gibbs et al., 2020*; *Hongo et al., 2007*; *Kane-maru et al., 2010*; *Khalili et al., 2016*; *Kronhed et al., 2009*; *Sen, Esmaeilzadeh & Eskiyurt, 2020*; *Stanghelle et al., 2020*; *Zhang et al., 2022a*; *Zhang et al., 2022b*.

Different tools were also adopted in these studies to measure outcome indicators, with SF-36 alone was used in six studies, Qualeffo-41 alone was used in three studies, OQLQ was used in two studies, SF-36 and Qualeffo-41 were used in two studies, and EQ5D and OQLQ were used in two studies. Overall, SF-36 was the most commonly utilized assessment tool, as it was employed in eight out of the 14 studies. SF-36 evaluates eight aspects of health-related quality of life (HRQOL), which belong to two major categories: physical component and mental component. Qualefo-41 and OQLQ are survey questionnaires specifically designed to evaluate the quality of life of patients with osteoporosis. The evaluation includes a total score and scores in five domains. EQ5D is a tool used to measure health-related quality of life, published by the European Quality of Life Research Group in 1990. The scale includes five health dimensions.

## Risk of bias

The results of the 14 included studies' bias risk assessment are shown in Fig. 2. All studies were classified as high risk or uncertain in terms of performance bias, eight studies were classified as low risk in terms of detection bias, 12 studies were classified as low risk in terms of follow-up bias and reporting bias, and 10 studies were classified as low risk in terms of other bias. Nine studies were considered as low risk in terms of random sequence generation or allocation concealment. The statistical proportion of each item in the bias risk assessment is shown in Fig. 3.

## Main outcome: general HRQOL

Eight studies in total evaluated the effects of exercise interventions on general HRQOL. The results of the meta-analysis of the experimental and control groups' combined effect size data on the general HRQOL are displayed in Fig. 4. After exercise interventions, the

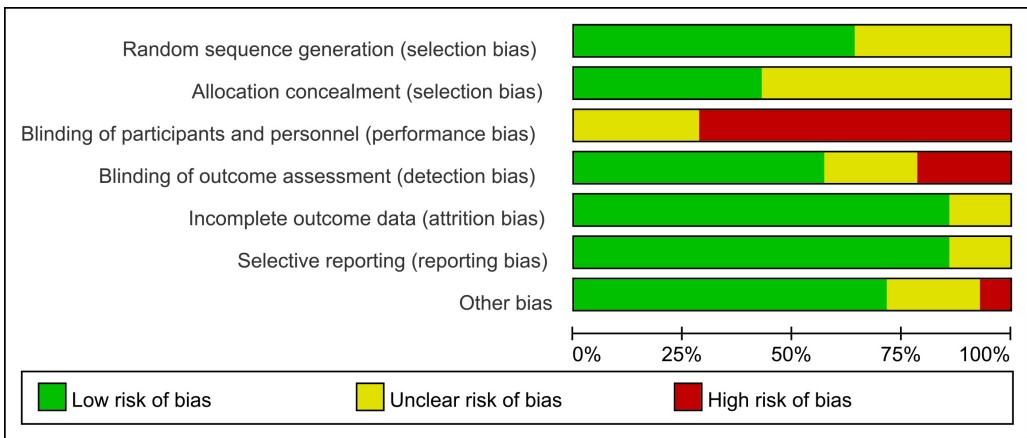

**Figure 3** Risk of bias graph: each risk of bias item is presented as percentages.

**Figure 4** Forest plot of the effects of exercises on general HRQOL. Note. *Çergel et al., 2019*; *Evstigneeva et al., 2016*; *Gibbs et al., 2020*; *Hongo et al., 2007*; *Kronhed et al., 2009*; *Sen, Esmaeilzadeh & Eskiyurt, 2020*; *Stanghelle et al., 2020*; *Zhang et al., 2022a*; *Zhang et al., 2022b*.

general HRQOL in the experimental group showed remarkable improvements compared to the control group (SMD = 0.79, 95% CI [0.34–1.24], $p = 0.0006$). Due to significant heterogeneity among the studies ($I^2 = 86\%$, $p < 0.00001$), sensitivity analysis was performed using the leave-one-out method, which indicated that the exclusion of any one study did not have a significant impact on the general HRQOL effect size (SMD = 0.68 to 0.92).

Subgroup analysis based on the exercise intervention programmes utilised in the eight trials was carried out in order to further examine the effects of varied exercise interventions on the improvements in overall HRQOL. Figure 5 displays the findings of the subgroup analysis based on exercise type, frequency, and duration. Resistance exercise (SMD = 1.01, 95% CI [0.50–1.52], $p = 0.0001$) significantly improved general HRQOL compared to multicomponent exercise (SMD = 0.58, 95% CI [−0.08–1.24], $p = 0.09$), according to the subgroup analysis based on exercise type. Exercise interventions that occurred three or more times per week (SMD = 0.80, 95% CI [0.22–1.38], $p = 0.007$) significantly improved general HRQOL compared to those that occurred less frequently (SMD = 0.79, 95% CI [−0.13–1.71], $p = 0.09$), according to the subgroup analysis based on exercise frequency.

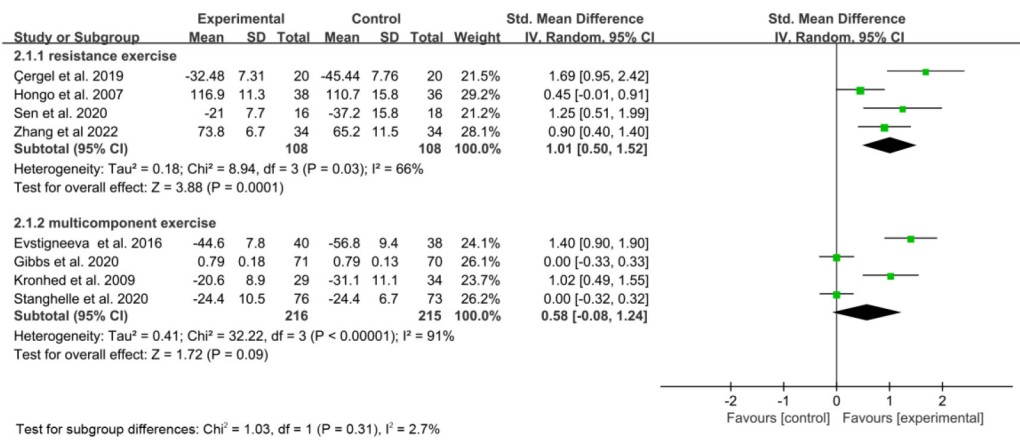

**(A)**

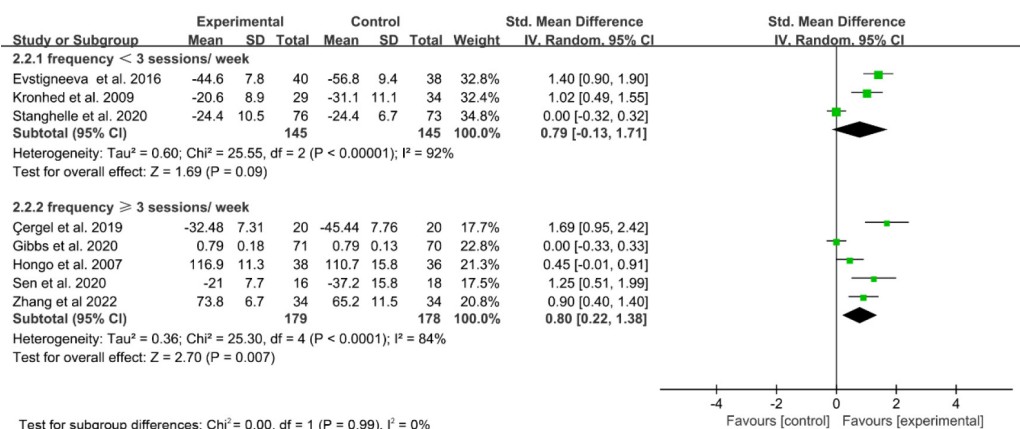

**(B)**

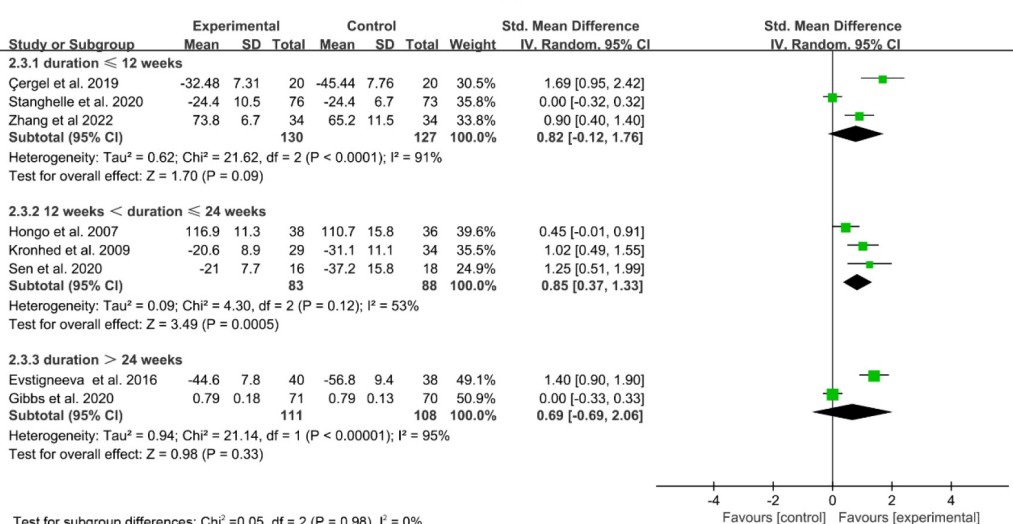

**(C)**

**Figure 5** **Subgroup analysis of general HRQOL by exercise mode.** (A) Description. (B) Frequency. (C) Duration. Note. *Çergel et al., 2019*; *Evstigneeva et al., 2016*; *Ferrara et al., 2019*; *Fu & Fan, 2021*; *Gibbs et al., 2020*; *Hongo et al., 2007*; *Kanemaru et al., 2010*; *Khalili et al., 2016*; *Kronhed et al., 2009*; *Sen, Esmaeilzadeh & Eskiyurt, 2020*; *Stanghelle et al., 2020*; *Zhang et al., 2022a*; *Zhang et al., 2022b*.

Also, the subgroup analysis based on exercise duration showed that exercise interventions lasting 13–24 weeks (SMD = 0.85, 95% CI [0.37–1.33], $p = 0.0005$) had a greater and statistically significant impact on general HRQOL than interventions lasting 12 weeks and below (SMD = 0.82, 95% CI [−0.12–1.76], $p = 0.09$) or more than 24 weeks (SMD = 0.69, 95% CI [−0.69–2.06], $p = 0.33$).

## Secondary outcome: physical HRQOL

Physical HRQOL includes bodily pain, physical function, role physical, and general health. Regarding the outcomes, bodily pain, physical function, role physical and general health were measured in 11, nine, eight and 11 studies, respectively. Figure 6 shows the results of the meta-analysis of the combined effect sizes data for each component of the physical HRQOL in the experimental and control groups. Combined results showed that the experimental group was superior to the control group in the reduction of bodily pain (SMD = 0.51, 95% CI [0.24–0.78], $p = 0.0002$), the improvement of physical function (SMD = 0.56, 95% CI [0.21–0.91], $p = 0.002$), physical (SMD = 0.39, 95% CI [0.14–0.64], $p = 0.003$), and general health (SMD = 0.68, 95% CI [0.25–1.11], $p = 0.002$). The results of the sensitivity analysis were similar to those presented above, with no outstanding differences.

## Secondary outcome: mental HRQOL

Mental HRQOL includes vitality, social function, role emotion, and mental health. Regarding the results, vitality, social function, role emotion and mental health were measured in 8, 10, 9 and 10 studies, respectively. Figure 7 shows the results of a meta-analysis of the combined effect sizes data for each component of the mental HRQOL in the experimental and control groups. The combined results demonstrated that the group that received exercise interventions improved vitality (SMD = 0.58, 95% CI [0.15–1.01], $p = 0.008$), social function (SMD = 0.37, 95% CI [0.17–0.58], $p = 0.0004$), and mental health (SMD = 0.50, 95% CI [0.25–0.74], $p < 0.0001$) in comparison to the control group. Despite being modestly favorable, the combined effect of role emotion was not statistically significant (SMD = 0.25, 95% CI [−0.03–0.54], $p = 0.08$). The results of the sensitivity analysis were similar to those presented above, but the differences were not significant.

## Publication bias

Because ten or more studies were included in the meta-analysis of the effects of exercise interventions on bodily pain, general health, social function, and mental health, we used a funnel plot to test their publication bias. From Fig. 8, we can see that the scatter is upward distributed, with a fundamental balance between left and right, and no significant publication bias among the studies.

## DISCUSSION

Previous studies have explored the effects of exercise on HRQOL in middle-aged and older osteoporosis patients by designing different forms of exercise interventions, but the conclusions obtained were not fully consistent. This study provided a systematic review

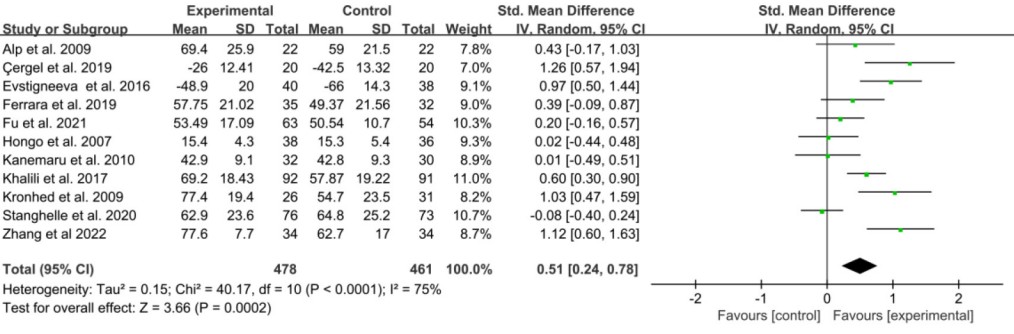

(A)

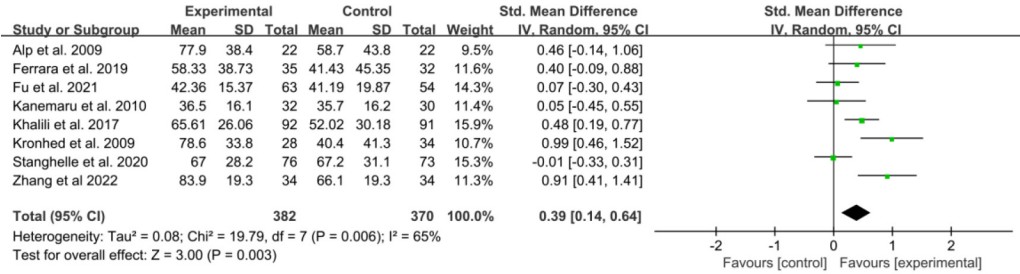

(B)

(C)

(D)

**Figure 6** **Forest plot of the effects of exercises on four physical HRQOL domains.** (A) Bodily pain. (B) Physical function. (C) Role physical. (D) General health. Note. *Alp et al., 2009*; *Arnold et al., 2008*; *Çergel et al., 2019*; *Evstigneeva et al., 2016*; *Ferrara et al., 2019*; *Fu & Fan, 2021*; *Gibbs et al., 2020*; *Hongo et al., 2007*; *Kanemaru et al., 2010*; *Khalili et al., 2016*; *Kronhed et al., 2009*; *Sen, Esmaeilzadeh & Eskiyurt, 2020*; *Stanghelle et al., 2020*; *Zhang et al., 2022a*; *Zhang et al., 2022b*.

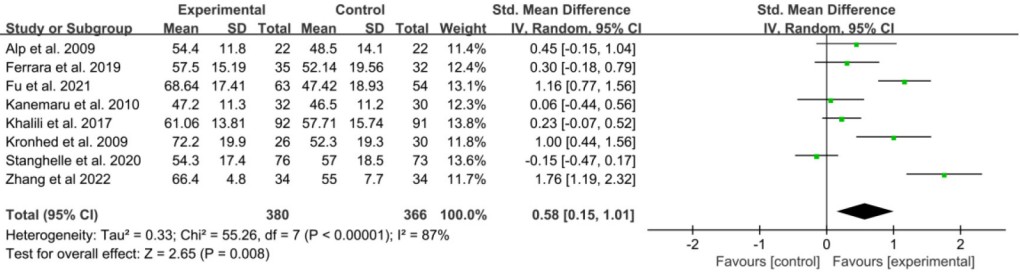

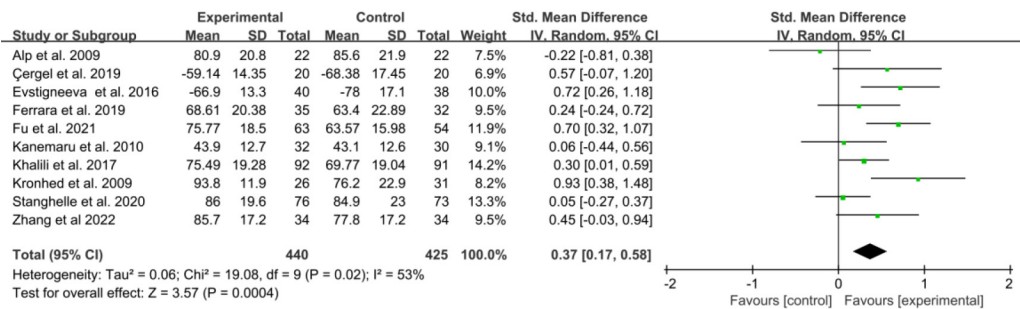

**Figure 7  Forest plot of the effects of exercises on four mental HRQOL domains.** (A) Vitality. (B) Social function. (C) Role emotion. (D) Mental health. Note. *Alp et al., 2009*; *Arnold et al., 2008*; *Çergel et al., 2019*; *Evstigneeva et al., 2016*; *Ferrara et al., 2019*; *Fu & Fan, 2021*; *Gibbs et al., 2020*; *Hongo et al., 2007*; *Kanemaru et al., 2010*; *Khalili et al., 2016*; *Kronhed et al., 2009*; *Sen, Esmaeilzadeh & Eskiyurt, 2020*; *Stanghelle et al., 2020*; *Zhang et al., 2022a*; *Zhang et al., 2022b*.

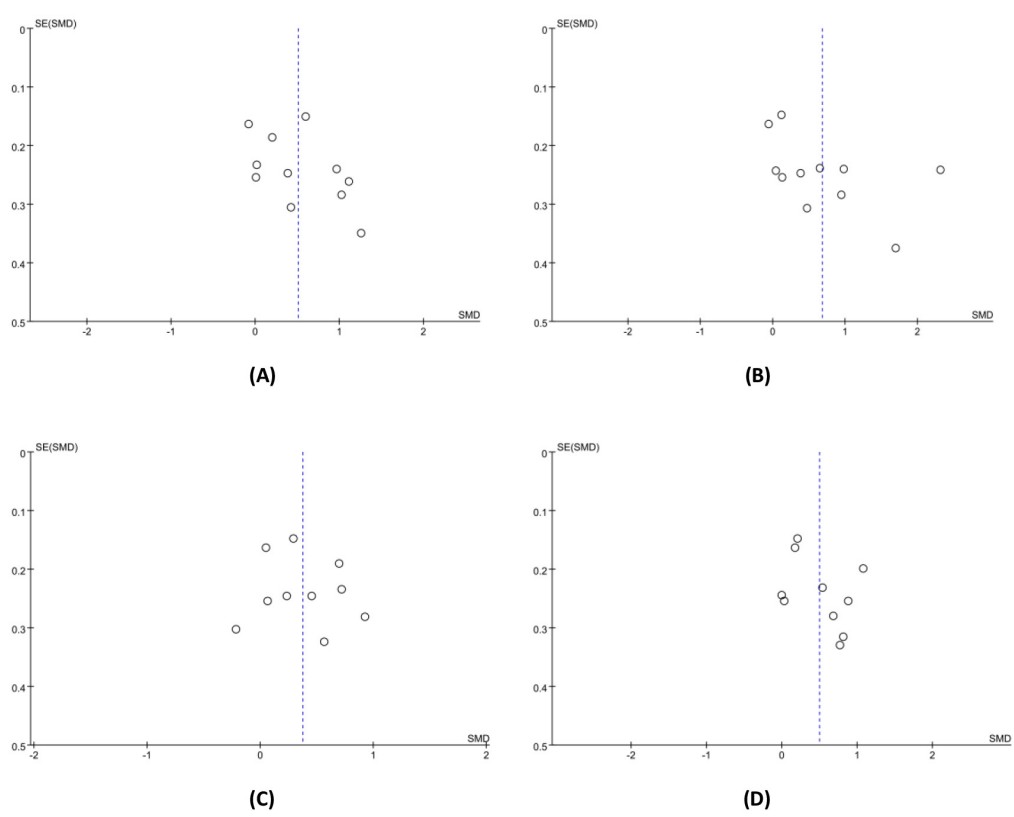

**Figure 8** **The funnel plot for publication bias.** (A) Bodily pain. (B) General health. (C) Social function. (D) Mental health.

and meta-analysis of randomized controlled trials with the highest level of evidence from the evidence-based medicine perspective. The results showed that exercise interventions had a significant positive effect on both general HRQOL and various sub-components of physical HRQOL. In the mental HRQOL domain, three components showed a significant positive correlation with exercise, with role emotion being an exception. The results of this study confirmed previous systematic reviews, suggesting that exercise interventions were able to help improve HRQOL in middle-aged and older patients with osteoporosis. The two previous reviews (*Anupama et al., 2020*; *Perez et al., 2021*) were only qualitative systematic evaluations and no meta-analysis was performed. Thus, our study provides stronger evidence to summarize and confirm the true significance of exercise interventions on HRQOL in middle-aged and older patients with osteoporosis.

In the included studies, the total effect of exercise interventions on general HRQOL was significant in middle-aged and older osteoporosis patients, but some studies did not support such a correlation. Specifically, two studies reported an effect size of zero (*Gibbs et al., 2020*; *Stanghelle et al., 2020*), and another study found a positive but non-significant effect size (*Hongo et al., 2007*). Moreover, the meta-analysis results also indicated substantial heterogeneity across studies. Among the four studies that contributed the most to the overall effect, three of them adopted resistance exercise. It is believed that these differences may

be due to the different exercise intervention programmes used in the studies included. The mechanism through which physical exercise influences HRQOL of middle-aged and older adults with osteoporosis remains incompletely understood. Potential contributing factors may include: the prevention of bone density loss and the provision of both short-term and long-term protective effects on bone health (*Ciaccioni et al., 2019*); the maintenance of adequate flexibility, strength, and interlimb coordination, which has a positive impact on the psychological health perception of older adults (*Ciaccioni et al., 2022*).

To explore the differences in the effects of exercise interventions on the general HRQOL of middle-aged and older osteoporosis patients, and to determine the relatively optimal exercise intervention plan, subgroup analysis was conducted based on intervention type, frequency, and duration. According to Cohen's recommended effect size evaluation criteria (*Cohen, 1988*), although multicomponent exercise had a small effect size ($d = 0.58$) on general HRQOL, it was not significant ($p = 0.09$). In comparison, resistance training had a large effect size ($d = 1.01$) and was statistically significant ($p = 0.0001$) in improving general HRQOL. Two previously published meta-analyses have shown that compared to multicomponent exercise, resistance training can better improve the balance ability of osteoporotic patients, thereby reducing the risk of falls and fractures (*Luan et al., 2019*; *Varahra et al., 2018*). Additionally, a systematic review has also shown that resistance training has a positive impact on the physical function and daily activities of older osteoporotic patients (*Wilhelm et al., 2012*). All of these studies may partly explain the large and significant correlation between the resistance training intervention and general HRQOL. Subgroup analyses on intervention frequency and duration showed that although both more and less than three times per week interventions achieved a moderate effect size, the latter did not have statistical significance. This has been in line with two other studies on the relation between exercise frequency and health in the older people, which discovered that at least three times per week's exercise may be critical for elder respondents' bone health and quality of life (*Kell & Rula, 2019*; *Kemmler & Von Stengel, 2013*). Additionally, we observed that exercise interventions lasting more than 24 weeks did not lead to greater improvements in general HRQOL. This may be because compliance of participants may decrease with prolonged intervention duration. According to *Sluijs, Kok & Vanderzee (1993)*, patients' compliance with exercise plans was a critical element affecting the success of exercise therapy. Therefore, in long-term exercise interventions, efforts should be made to strengthen patient compliance motivation. Lastly, we compared the results of the subgroup analysis with the recommendations of the WHO Guidelines on Physical Activity and Sedentary Behavior. A notable consistency in the types and frequency of interventions has been found out. The guidelines recommend that older adults engage in strength training exercises targeting major muscle groups more than twice a week as part of their weekly physical activity, due to the additional health benefits these exercises can provide (*World Health Organization, 2020*).

All four subcomponents in the domain of physical HRQOL showed significant improvement, which was striking as this result was not entirely consistent with the evidence reported in previous reviews and meta-analysis. According to *Li et al. (2009)* meta-analysis, exercise only significantly reduced bodily pain and improved physical

function in osteoporosis patients. These differences may be due to the measurement questionnaires used in the included studies. Of the five studies included in *Li et al. (2009)* analysis, three used Qualeffo, while the studies we included mostly used SF-36. The study by *Lips et al. (1999)* appeared to support this hypothesis by comparing the differences between the Qualeffo and SF-36 scales in measuring HRQOL scores in osteoporosis patients. The results indicated a significant correlation between the two scales in terms of the bodily pain and physical function scores; the use of the SF-36 scale tended more discriminative in terms of general health awareness scores.

For mental HRQOL, we found that exercise interventions significantly improved vitality, mental health, and social function, with the first two having relatively large effects. According to prior research, physical exercise significantly improved sleep, energy levels, and reduced negative emotions, despair and anxiety for a substantial number of middle-aged and older people (*Myers et al., 1999*; *Witard et al., 2016*). From a physiological point of view, mediators between exercise and mental health include metabolic changes in central neurotransmitters such as serotonin and endogenous opioids, as well as sleep regulation (*Stathopoulou et al., 2006*). Our results confirmed a significant positive association between exercise interventions and vitality, between exercise interventions and mental health in the special population of middle-aged and older osteoporosis patients. However, role emotion examined role limitations caused by emotional issues, which were related to personality factors, with personality being a relatively stable trait (*Wang et al., 2007*). This may explain why, in our study, although the role emotion was improved, it was not statistically significant.

Despite a lower incidence rate in men compared to women, osteoporosis remains a significant health issue for the former (*Choi, Lee & Lee, 2021*). Our review revealed that the majority of participants in the included studies were women, comprising 94.81% of the total sample size. This gender imbalance highlights the need for future research to focus on the effects of exercise interventions on middle-aged and older men with osteoporosis. Furthermore, exercise intensity is a critical moderating variable, influencing individuals' perceptions of HRQOL (*Knowles et al., 2015*). However, the systematic review in this research did not account for exercise intensity due to the lack of detailed information provided by most studies. Future research should monitor exercise intensity adopting measures such as heart rate reserve, maximum heart rate, maximum oxygen uptake, or rating of perceived exertion scales to establish evidence for optimal exercise intensity.

In our study, it needed to be acknowledged that there were still some limitations that could not be ignored. Firstly, methodological flaws in the research included may have exaggerated the current results, mainly due to the fact that participants in most studies have not been blinded, leading to a high risk of implementation bias. Secondly, it was necessary to study the differences in the impact of exercise interventions on HRQOL based on patient characteristics such as age, gender, and history of fractures, but the characteristics of the studies included did not allow for this. Thirdly, due to the use of different effect size indicators in the studies included, mean and standard deviation of the final measurement results were estimated based on formulas to avoid data loss, and the combined effect size were calculated accordingly, which may have led to bias to certain degree. Additionally,

the meta-analyses for some of the outcomes had a substantial statistical heterogeneity, therefore our results should be interpreted with caution.

## CONCLUSION

This study summarized the existing evidence from randomized controlled trials on the relationship between exercise interventions and HRQOL of middle-aged and older osteoporosis patients. The results of the meta-analysis showed that exercise interventions had a significant positive effect on HRQOL of middle-aged and older patients with osteoporosis. Resistance training had a more pronounced effect on improving HRQOL compared to multicomponent exercise. The frequency of exercise intervention at three times or more per week proved to lead to a more significant improvement in HRQOL. Exercise intervention lasting for 13–24 weeks had a more significant effect on HRQOL improvement compared to other durations. This finding has significant clinical and academic implications for improving the HRQOL of middle-aged and older adults with osteoporosis. It not only provides evidence-based support for the formulation of exercise prescriptions and health management policies, but also offers references for other researchers engaged in this field.

### Funding
The authors received no funding for this work.

### Competing Interests
The authors declare there are no competing interests.

### Author Contributions
- Di Geng conceived and designed the experiments, performed the experiments, analyzed the data, prepared figures and/or tables, authored or reviewed drafts of the article, and approved the final draft.
- Xiaogang Li conceived and designed the experiments, performed the experiments, analyzed the data, prepared figures and/or tables, authored or reviewed drafts of the article, and approved the final draft.
- Yan Shi conceived and designed the experiments, performed the experiments, authored or reviewed drafts of the article, and approved the final draft.

### Data Availability
This is a systematic review and meta-analysis.

### Supplemental Information
Supplemental information for this article can be found online at http://dx.doi.org/10.7717/peerj.18889#supplemental-information.

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
