# Peer review of "Effect of exercise intervention on health-related quality of life in middle-aged and older people with osteoporosis: a systematic review and meta-analysis"

_PeerJ, doi:10.7717/peerj.18889_

## Round 0.1 · original submission · Minor Revisions

Thank you for your submission.

The reviewers have provided feedback on your work and suggested that minor revisions are required. Whilst some comments appear minor, there may be more substantial modifications required as a result of these changes.

A thorough review of spelling and grammar is needed along with including more information and rationale for the search strategy and study timeframes for inclusion.

There is more detail required in the description of exercise programs and outcome measures.

Reviewer 1 ·

Basic reporting

Basic reporting is adequate and figures are good. A good background on osteoporosis and quality of health and exercise was presented, but there were some grammatical errors in the manuscript, although it is still understandable and easy to fix. There were also some awkward phrases that could be rephrased (i.e., in Line 46, maybe consider rephrasing “second biggest killer merely next to…” to “second leading cause of death behind…”). And in Line 78, define what “…an equivalent amount of vigorous activity” is (vigorous activity is usually half the amount of moderate-intensity physical activity, but please clearly state that).

Experimental design

Yes, all of these criteria appear to be the case. But in Line 199, where it states, “… after which 798 more articles that did not meet the requirements of this study were excluded.” Please (re)state what these requirements were, in which these 798 articles lacked. You did mention the criteria that narrowed down 64 articles down to 14 articles in Lines 201 to 204, but please also do so when you went from 859 to 798.

Validity of the findings

All of these criteria also appear to be the case. Out of the different modalities of exercise investigated, resistance training and strength training appeared to have the most profound effect on health-related quality of life. However, more description and details of what types of resistance training and how it was programmed can be provided (e.g., weight training, weight machines, resistance bands, aqua resistance training, calisthenics/bodyweight exercises, etc.; and how many sets, repetitions, percent 1-repetition maximum (% 1-RM), etc.). This can help the reader know what types of resistance training to prescribe to patients with osteoporosis, in order to improve their quality of life.

Additional comments

While there are many studies on osteoporosis and quantity of life (e.g., mortality), there are few studies on osteoporosis and quality of life. And while there are many studies on the effects of exercise on osteoporosis, few investigate quality of life, which makes this meta-analysis very important and useful.

Reviewer 2 ·

Basic reporting

Dear colleagues,

1. Congratulations on this achievement. For me the topic is indeed compelling, and the title is well-aligned with the study's content, providing a clear and relevant focus. However, I reviewed the document and noted a few areas that could benefit from improvement.

2. To enhance the robustness of your study, it would be beneficial to incorporate additional citations that support the rationale behind your research. Expanding the reference base will not only strengthen the theoretical foundation of your study but also provide a more comprehensive context, showcasing alignment with existing literature and demonstrating thorough engagement with the topic.
I noticed quite a few very old citations in the text, but I put this as an observation, they should not be removed, I recommend you cite some other works in the same direction from the more recent specialized literature

3. It seems that the study does not include a clear hypothesis. A hypothesis should serve as the foundation of your research, outlining where the study begins and the goals you aim to achieve. Without it, the study lacks direction, which makes it difficult to assess its value or conclusions.

4. In the conclusions, the proposed hypothesis must be confirmed or refuted, with the necessary objective values ​​or indices.

5. Limitations of the study must be introduced.

6. Perspectives.

Experimental design

no comment

Validity of the findings

no comment

Additional comments

no comment

·

Basic reporting

No comments

Experimental design

Kindly address the following:

1. Did the search include unpublished studies such as trial registries?
2. What was the level of agreement between the researchers responsible for selecting the primary research studies? It would be great if you could provide the kappa statistic.
3. Did the researchers do a power calculation and sample size estimation?
4. Kindly provide the rationale for considering exercise duration of 12 weeks and less, 13 -24 weeks and more than 24 weeks?

Validity of the findings

Kindly address the following:

1. The level of heterogeneity is high as seen in the results. How does it influence your results?
2. Does the trial have enough power to consider the results of the subgroup analyses as valid?

Additional comments

Congratulations to the authors on a well conducted research study. The topic chosen also deserves appreciation.

---

## Round 0.2 · accepted · Accept

Thank you for your efforts in addressing the reviewers' feedback. I am pleased you have addressed the relevant comments and made the necessary changes for publication.

Reviewer 1 ·

Basic reporting

Authors addressed reviewer's comments on basic reporting in the revised manuscript.

Experimental design

Authors addressed reviewer's comments on experimental design in the revised manuscript.

Validity of the findings

Authors addressed reviewer's comments on validity of findings in the revised manuscript.

Additional comments

The revised manuscript is an improvement from the original manuscript. The English is also much improved. There were just a few grammatical errors in the manuscript, which I trust will be corrected when making and reviewing the proofs.

·

Basic reporting

no comment

Experimental design

Authors have properly addressed the queries.

Validity of the findings

The queries raised were sufficiently addressed by the authors.

Additional comments

Thank you for the responses.